# Distribution and Genetic Diversity of the Amphibian Chytrid in Japan

**DOI:** 10.3390/jof7070522

**Published:** 2021-06-29

**Authors:** Koichi Goka, Jun Yokoyama, Atsushi Tominaga

**Affiliations:** 1National Institute for Environmental Studies, 16-2 Onogawa, Tsukuba 305-8506, Japan; 2Department of Biology, Faculty of Sciences, Yamagata University, 1-4-12 Kojirakawa, Yamagata-shi, Yamagata 990-8560, Japan; jyokoyam@sci.kj.yamagata-u.ac.jp; 3Department of Natural Sciences, Faculty of Education, University of the Ryukyus, 1 Senbaru, Nishihara, Okinawa 901-0213, Japan; tominaga@edu.u-ryukyu.ac.jp

**Keywords:** chytrid, Batrachochytrium dendrobatidis, nested PCR, ITS, Cynops ensicauda

## Abstract

While research on frog chytrid fungus *Batrachochytrium dendrobatidis (Bd)*, an infectious disease that threatens amphibian diversity, continues to advance worldwide, little progress has been made in Japan since around 2010. The reason for this is, which we pointed out in 2009, that the origin of frog chytrid fungus may be in the East Asian region, including Japan based on the *Bd* ITS-DNA variation, and as few cases of mass mortality caused by this fungus have been observed in wild amphibian populations in Japan, the interest of the Japanese government and the general public in *Bd* has waned. However, we believe that organizing the data obtained so far in Japan and distributing the status of frog chytrid fungus in Japan to the world will provide useful insight for future risk management of this pathogen. We collected more than 5500 swab samples from wild amphibians throughout Japan from 2009 to 2010. Then, we investigated the infection status using the Nested-PCR method. We sequenced the obtained DNA samples and constructed a maximum-parsimony (MP) tree to clarify the phylogenetic diversity of *Bd*. We detected *Bd* infection in 11 (nine native and two alien) amphibian species in Japan and obtained 44 haplotypes of *Bd* ITS-DNA. The MP tree showed a high diversity of *Bd* strains in Japan, *suggesting* that some strains belong to *Bd*-GPL and *Bd*-Brazil. Except for local populations of the Japanese giant salamanders *Andrias japonicus* in Honshu Island and the sword tail newts *Cynops ensicauda* in Okinawa Island, the *Bd* infection prevalence in native amphibian species was very low. The alien bullfrog *Aquarana catesbeiana* had high *Bd* infection rates in all areas where they were sampled. No *Bd* infection was detected in other native amphibians in the areas where giant salamanders, sword tail newts, and bullfrogs were collected, suggesting that many native amphibians are resistant to *Bd* infection. The sword tail newt of Okinawa Island had both the highest infectious incidence and greatest number of haplotypes. The giant salamanders also showed relatively high infection prevalence, but the infected strains were limited to those specific to this species. These two Caudata species are endemic to a limited area of Japan, and it was thought that they may have been refugia for *Bd*, which had been distributed in Japan Islands for a long time.

## 1. Introduction

The global pandemic of a disease of amphibians caused by the chytrid fungus *Batrachochytrium dendrobatidis* is a serious threat to the conservation of biodiversity. Since its discovery by Berger et al. [1], chytridiomycosis due to *B. dendrobatidis* (*Bd*) has been proposed as the infectious disease responsible for declines in wild frog populations in Mesoamerica, South America, and Oceania countries [1,2,3,4,5,6,7,8,9].

Because of the importance of the problem, clarifying the geographic source of *Bd* and the mechanism of its spread has become the focus of intense research. Until now, two hypotheses have been proposed to account for the emerging nature of *Bd* around the world. The first is the “novel pathogen hypothesis”, which states that the spread of *Bd* into new geographical areas by global trade of amphibians [1,10,11]. This hypothesis has been supported by the evidence of low genetic variation *Bd* around the world or reduction of genetic diversity of in *Bd* from recently infected amphibian populations, based on the data of multilocus sequence typing or microsatellite analysis [12,13,14,15].

The second is the “endemic pathogen hypothesis”, which states that the emergence of chytridiomycosis has been caused by amphibian hosts becoming more susceptible to pre-existing infections as a consequence of environmental changes [16,17,18,19]. This hypothesis has received some support from data showing that *Bd* presented in global amphibian populations many decades ago: by 1933 in Cameroon [20], 1934 in Kenya [21], 1938 in South Africa [8], 1961 in Canada [19], 1894 in Brazil [22], 1888 in USA [23], and 1894–1929 in Mexico [24].

Subsequently, Farrer et al. [25], Schloegel et al. [26], and Rosenblum et al. [27] sequenced whole genome of *Bd* and discovered that at least three major lineages of *Bd* exist in the world; the Cape lineage (*Bd*-Cape), the Swiss lineage (*Bd*-CH), the Brazil lineage (*Bd*-Brazil), other than the Global Pandemic lineage *(Bd*-GPL), which has been considered the pathogen of worldwide amphibian decline.

Moreover, later, using phylogenetic analysis of a complete DNA dataset of 234 *Bd* isolates collected and cultured over nearly 20 years, O’Hanlon et al. [28] showed that the Korean *Bd* lineage, *Bd*-ASIA-1, shows ancestral relationships with other lineages, and that the Bd-ASIA-1 lineage shares more diversity with the global *Bd* population than any other lineage, suggesting an East Asian origin for *Bd*.

Furthermore, in a recent study by Byrne et al. [29], multilocus genotyping for swab samples confirmed the results of O’Hanlon et al. [28] and identified a new lineage, *Bd*-ASIA-3, found in East Asia (Philippines, Indonesia, and China). The complete absence of lethal outbreaks or population declines due to chytridiomycosis in Asia, despite the widespread occurrence of *Bd*, was considered evidence of the endemic nature of host-pathogen interactions [30].

On the other hand, Byrne et al. [29] pointed out that their own method that select samples for genotyping based on the positive results of qPCR may introduce sampling bias and detect only *Bd*-GPL genotypes. Therefore, the currently estimated diversity of *Bd* may still be greatly underestimated, suggesting that many undiscovered *Bd* strains are still latent, especially in Asia where the number of samples is small.

In Japan, Asia’s first case of chytridiomycosis was confirmed in December 2006 in an exotic frog, *Ceratophrys ornata*, which was being kept in captivity [31]. Then, it became a major topic of discussion that endemic amphibians might become extinct due to a pandemic of this disease, and many researchers began to investigate. From 2007 to 2008, our research team promoted a nationwide survey and analyzed the genetic variation of *Bd* using the ITS-DNA region. The results showed that the Japanese endemic giant salamander is infected by a specific lineage and that the Okinawa Island newts possessed the most diverse lineage in the world at that time [32].

At that time, there was very little information on frog chytrid fungus in the Asian region, so our findings highlight the importance of the Asian region in the search for the origin of *Bd*. However, as our results indicated that *Bd* was likely to be an endemic pathogen in Japan, and as there was little evidence of damage to native species in Japan, the sense of urgency about *Bd* diminished within the Ministry of the Environment and among many researchers in Japan. Our research team also had to deal with more important invasive alien species such as fire ants, and the frog chytrid research project was discontinued.

On the other hand, researchers all over the world are focusing on the diversity and endemism of *Bd* in East and Southeast Asia, where phylogenetic studies have been lagging behind. Naturally, the *Bd* of the Japanese archipelago is also an important subject of investigation for clarifying the origin and diversity of *Bd* in Asia and the world.

We have accumulated the ITS-DNA sequences of *Bd* from all around the Japan Islands as a continuance of our previous study in order to grasp the *Bd* infection state in wild amphibians since 2009 to 2010. The only data we have are ITS-DNA sequences of frog chytrid fungi collected from various parts of Japan, which are very short sequences for phylogenetic analysis of *Bd*, but we believe that they are useful for estimating the diversity of *Bd* lineages in Japan. In particular, there are very few data for the Japanese archipelago, even in the global-scale *Bd* diversity studies mentioned above.

In this paper, we analyze the data we have ever obtained to summarize the infection prevalence and the diversity of *Bd* lineages in various amphibians from Japan.

## 2. Materials and Methods

### 2.1. Swab Samples from Wild Amphibians in Japan

As in our previous study [32], we collected swab samples from various species at a range of sites in all prefectures of Japan’s four main islands (Hokkaido, Honshu, Shikoku, and Kyushu Island) and from Japan’s southwestern islands (Amami, Okinawa, Ishigaki, and Iriomote).

We surveyed a total of 950 collection sites, adding 14 new sites to our previous collection sites [32], and obtained 5517 new samples, representing 47 native species and 2 alien species. The landscape of the sampling points varied from agricultural fields to native forests. Amphibians were captured between June 2009 and July 2010 and were handled individually by using a new pair of disposable plastic gloves for each specimen to avoid contamination of samples. Immediately after we captured the wild specimens, we collected fungal samples by swabbing the ventral surface, legs, and feet of each amphibian with a sterile cotton swab (Men-tip 1P1501, Nihon-Menbo Co., Tokyo, Japan). We then released each amphibian at the sampling point. All swab samples were stored at −28 °C until analysis. *Batrachochytrium dendrobatidis* DNA was extracted from swab samples according to the same procedure used previously [32].

### 2.2. Nested PCR Assay

We used the nested PCR assay developed in our previous study [32] to detect *Bd*. A region of the 5.8S ribosomal RNA gene as well as the ITS1 and ITS2 regions were amplified by using a nested PCR assay involving two pairs of primers. During the first-round PCR amplifications, the outer primers *Bd*18SF1 (5′-TTTGTACACACCGCCCGTCGC-3′), and *Bd*28SR1 (5′-ATATGCTTAAGTTCAGCGGG-3′) amplified the DNA fragment between the end of the 18S rRNA gene and the start of the 28S rRNA gene; DNA extracted from swab samples was used as template. During the second-round amplification, the target ITS1–5.8S–ITS2 regions were amplified from the first-round PCR products by using the primers *Bd*1a (5′-CAGTGTGCCATATGTCACG-3′) and *Bd*2a (5′-CATGGTTCATATCTGTCCAG-3′), both of which were designed to be specific for *Bd* [33]. Thus, we obtained sufficient specific and highly concentrated PCR product to sequence of the amplified DNA fragments.

PCR assays were conducted according to the same procedure described previously [32]. Each product of the second-round amplification was sub-cloned into a vector plasmid by using a pT7 Blue Perfectly Blunt Cloning Kit (Novagen, EMD Bioscience, San Diego, CA, USA) and transferred into *Escherichia coli* in accordance with the manufacturer’s protocol. The cloned fragments in 3 positive clones for each nested-PCR product were sequenced by using the T7 promoter and U19 reverse primers and an ABI3730 Sequencer (Applied Biosystems, Thermo Fisher Scientific, Waltham, MA, USA).

### 2.3. Phylogenetic Analysis of the ITS Gene

To clarify the genetic divergence of *Bd* in Japan, we constructed a phylogenetic tree by using the ITS-DNA haplotypes detected in the present study and in our previous study [32]. For this analysis, we included *Bd* ITS-DNA sequences obtained from other countries: China [34], North and South America [26,32], and Italy [32], as the lineage comparators. We used ITS-DNA sequences from 6 other species of chytrids (AUS_3, ITA2590, AUS_8, AUS_9, AUS12, ITA2633, and ITA2580; see Goka et al. [32]) as outgroups. The tree was constructed in the maximum-parsimony method, as done previously [32].

We aligned *Bd* sequences with those from the selected outgroups by using ClustalX [35]. The alignment results were modified manually to minimize the numbers of insertions and deletions (indels). The phylogenetic relationship of ITS haplotypes of *B. dendrobatidis* was analyzed by using the maximum-parsimony method. We used PAUP* 4.0b [36] to reconstruct trees. All characters were weighted equally; all indels were coded as binary data (sequence present = 1, sequence absent = 0). The maximum-parsimony analysis was conducted through a heuristic search with the TBR branch-swapping option. To identify multiple islands of equally most-parsimonious trees, 100 rounds of random sequence additions were performed [37]. Bootstrap analysis [38] with 10,000 replications was performed by using the same program and analysis setting.

## 3. Results

### 3.1. Infection Status and Haplotype Variation of Bd in Japan

The present survey identified 207 infected wild amphibians among the 5517 sampled, leading to an incidence of infection of 0.038, which is similar to that of our previous study (0.041 (87 of 2103 animals)). We detected *Bd* on all of the islands we surveyed except Hokkaido, Ishigaki, and Iriomote (Figure 1).

We detected *Bd* infection in 11 amphibian species (Table 1), adding 3 native species to the results of Goka et al. [32]. Nine species were native, and the remaining 2 were alien (*Aquarana catesbeiana* and *Xenopus laevis*). The giant salamander (*A. japonicus*) of Honshu Island and the sword-tail newt (*Cynops ensicauda*) of Okinawa Island showed the highest prevalence of infection in both our previous [32] and the current (23 of 46 (0.500) and 87 of 137 (0.635), respectively) study. Notably, the *C. ensicauda* population in Amami Island showed lower prevalence (6 of 134 samples (0.045)).

The prevalence of *Bd* infection of all other native amphibian species was low (Table 1). The 4 native species (*Fejervarya limnocharis*, *Dryophytes japonicus*, *Pelophylax nigromaculatus*, and *Glandirana rugosa*) showing low prevalence in our previous study [32] again demonstrated low prevalence in the present study (6 of 987 (0.006), 1 of 700 (0.001), 2 of 656 (0.003), and 2 of 346 (0.006), respectively) despite an increase in the number of samples.

In addition, among the species newly identified as *Bd-*infected, *Pelophylax porosus* from Honshu Island and *Buergeria japonica* from Amami Island had remarkably low prevalence of *Bd* infection (3 of 188 (0.016) and 1 of 71 (0.014), respectively). In contrast, the new *Bd* host, *O. narina* of Okinawa Island, had a relative high prevalence of *Bd* infection (0.091), but the sample size (*n* = 11) was small. The two naturalized alien species—*A. catesbeiana* and *X. laevis*—showed relatively high prevalence values (approximately 0.3).

In the present study, we detected 44 haplotypes (A through L, N through P, S, T, W, Y, Z, and *Bd*27 through *Bd*50) from wild amphibians (Figure 1). Of these 44 haplotypes, 24 (*Bd*27 through *Bd*50; GenBank accession numbers, AB723963 through AB723986) were discovered during the present study. The native species had 33 haplotypes, whereas alien species carried 17 haplotypes.

The most frequently detected haplotype was A (94 of 207 samples (0.454)). Except for the K haplotype, which is unique to the giant salamander, the next most frequent haplotypes were E (13 of 207 (0.063)) and C (7 of 207 (0.034)). Each of the other 41 haplotypes was detected in only 1 or a few animals (Figure 1).

In mainland Japan, the amphibian species carrying highest diversity of *Bd* haplotypes was the American bullfrog (*A. catesbeiana*), which carried 17 haplotypes. However, the sword-tail newt (*C. ensicauda*) on Okinawa Island had much higher *Bd* haplotype diversity, carrying as many as 28 haplotypes.

Almost all infected animals carried only 1 haplotype. Among 207 samples, only 8 *A. catesbeiana* and 2 *C. ensicauda popei* carried 2 haplotypes each. Therefore only 5% of infected samples showed polymorphic ITS-DNA of *Bd*.

### 3.2. Phylogenetic Analysis of ITS-DNA Haplotypes in Bd

We inferred 37 indels from the results of the multiple alignment and included these in the phylogenetic analysis as binary data. Among a total of 359 characters, 35 (including 25 indels) were parsimony-informative among *Bd* accessions. We obtained 6164 most parsimonious (MP) trees of 298 steps in a single island, at a CI of 0.79 (CI excluding uninformative characters, 0.74) and an RI of 0.91.

As a result, four major groups were recognized in all MP trees (Figure 2). One (Group I) comprised three haplotypes (B, J, and K) with a high bootstrap value (99%), and the other was composed of all remaining haplotypes. The result that *Bd* lineage specific to the giant salamander constitutes a unique clade is consistent with our previous study.

This second group was split into three subgroups in all MP trees: two (groups II and III) were formed by 16 Japanese haplotypes and haplotypes from other countries that were assigned to *Bd*-Brazil [26], and the remaining subgroup (group IV) comprised all remaining Japanese haplotypes and all haplotypes from other areas and assigned to *Bd*-GPL [26]. Thus, the topology of the MP phylogenetic tree suggested that the Japanese *Bd* lineage encompasses the globally distributed lineages *Bd*-GPL and *Bd*-Brazil. The sward-tailed newts (*C. ensicauda*) of Okinawa Island carried *Bd* strains belonging to Groups II, III, and IV. On the other hand, the sward-tailed newts of Amami Island and other native amphibian species except of the giant salamander carried only Group IV *Bd* strains. The alien bullfrogs (*A. catesbeiana*) also carried strains belonging to Groups II, III, and IV (Figure 1).

## 4. Discussion

Adding the six haplotypes (M, Q, R, U, V, and X) that have been detected only in amphibians bred in pet shops and institutions [32] to the haplotypes detected in the present study, we have identified 50 haplotypes of *Bd* in Japan to date. Interestingly, most of the infected amphibians carried a single *Bd* haplotype, as same as the result of our previous study [32], and only 5% of infected amphibian samples showed polymorphic (two haplotypes) ITS-DNA of *Bd*. In contrast, Bai et al. [34] found much higher heterogeneity of *Bd* in individual amphibians in China. In their study, 26 of 66 infected samples each carried a single haplotype, and 30, seven, and three animals carried two, three, and four haplotypes, respectively, indicating that as many as 60.6% of infected amphibians carried multiple *Bd* haplotypes in China. Such a difference in the number of *Bd* haplotypes infecting a single amphibian between Japan and China is of great interest and may become a key factor for revealing the co-evolutionary history between amphibians and *Bd*. Perhaps *Bd* was introduced into China recently, allowing multiple haplotypes to infect a single amphibian simultaneously; *Bd* in Japan may have inhabited each amphibian strain or population for a long time, resulting in a one-to-one relationship between host and *Bd* races. Although this scenario is of course only one possibility, investigating the genotypic relationships between host and *Bd* in various host species and geographic regions merits attention.

Based on the MP tree of the *Bd* ITS-DNA haplotype, the lineages infecting the giant salamander *A. japonicus* form a phylogenetically unique group, suggesting a special relationship with the host giant salamander. In particular, the *Bd* of the Japanese giant salamander was detected in a specimen from 1902 [32], which has been the oldest case of *Bd* infection in Japan. The ITS-DNA of the same sequence as that of the Japanese giant salamander-specific *Bd* lineage has been reported from one frog species native to China [34], but no other case has been reported so far. This giant salamander-specific *Bd* strain is expected to be a key in investigation of the origin and evolution of the chytrid fungus in Asia. In the future, we should establish an isolation culture of this strain and analyze the detailed genetic and pathological information.

Three other phylogenetic groups were divided in the MP-tree, two of which were presumed to be the *Bd*-Brazil lineage and the other one the *Bd*-GPL lineage, suggesting that the Japanese *Bd* lineage encompasses the globally distributed lineages *Bd*-GPL and *Bd*-Brazil. All *Bd* strains carried by native Japanese amphibians other than the giant salamander and the sward-tailed newt belong to the *Bd*-GPL strain. On the other hand, the sward-tailed newt from Okinawa Island were infected with both *Bd*-Brazil and *Bd*-GPL strains. The *Bd*-GPL strain has been reported worldwide, but *Bd*-Brazil has only been reported from the North America, Korea, and Brazil so far. Moreover, *Bd*-Brazil has been found in native amphibians only in Brazil [26,28,39,40]. Our results indicate that *Bd*-Brazil has infected an isolated endemic amphibian, the sward-tailed newt *C. ensicauda*, on a remote Japanese island, that would be the first case of native species with *Bd*-Brazil outside Brazil.

Furthermore, the sward-tailed newt of Okinawa Island had the highest *Bd* infection rate in Japan and was the host infected with the most diverse *Bd* strains, meaning that this species is the core of *Bd* diversity in Japan.

These results strongly suggest that sward-tailed newt in the Okinawa Island is a very important species for exploring the origin and history of genetic differentiation of *Bd* not only in Asia but also worldwide. The earliest data on *Bd* traces in Brazil suggest that *Bd* was invading or inhabiting the country as early as 1894, and it appears that the early invasion was by *Bd*-Brazil, followed by a secondary introduction of *Bd-*GPL into Brazil in the 1970s [22,40]. The first Japanese immigration from Japan to Brazil is officially recorded to have been in 1908, and even before that, a group of Japanese immigrants in Hawaii is considered to have gone to South America [41,42]. It is possible to envision a scenario in which these historical flows of people and trade led to the *Bd*-Brazil spillover from Japan. In order to explore the history of *Bd* in Japan and around the world, accumulating detailed genetic information on *Bd* itself as well as tracing the history of international immigration and trade may provide important insights.

Interestingly, the incidence of infection in the sword-tail newt population of Amami Island, was quite low (six of 134 animals (0.045)), compared to those in the newt population of Okinawa Island. The two island populations of the sward-tailed newt have been considered to have a sub-species relationship [43]. Later, Tominaga et al. [44] showed the presence of remarkable genetic differentiation between the Okinawa and Amami Islands populations of the newt based on phylogenetic analysis of the mitochondrial cytochrome b gene, supporting the validity of the subspecies relationship between them. Moreover, recently, The Herpetological Society of Japan named the Amami and Okinawa Islands populations of the sward-tailed newt as subspecies; *Cynops ensicauda ensicauda* and C. ensicauda popei, respectively (see http://herpetology.jp/wamei/index_j.php (accessed on 29 June 2021)).

These sword-tail newt sub-species have been separated for 3 to 5 million years, according to molecular phylogeny by Tominaga et al. [44]. Such genetic differentiation likely affected the evolution of resistance or immunity against *Bd*. Ecologic, physiologic, and environmental characteristics may be the keys to elucidating the mechanisms of differentiation in *Bd* infection between these two sub-species of sword-tailed newts.

In addition, only a few amphibian species other than newts were found to be infected in the entire Southwestern Islands, and it is necessary to increase the number of samples to investigate the infection status in these islands in detail. The fauna of the Southeast Islands of Japan is characterized by a high ratio of endemic taxa and genetically diverged populations, most of which have supposedly been isolated from their relatives as a result of island formation, e.g., [45,46,47]. These islands are also likely to be important research points for exploring the diversity of *Bd* fungus.

On the other hand, the exotic bullfrog had a high infection rate in all areas where it was captured. This result is consistent with previous foreign studies [39,48] that suggest that bullfrogs act as boosters of *Bd* and contribute to the spread of *Bd* throughout the world [26,29,40,49,50].

The infection rate of native amphibians was low, even in the vicinity where bullfrogs live and on the Okinawa Island where the highly infected sword-tailed newts inhabit, suggesting that native amphibians have some mechanisms of resistance to *Bd* infection as a lot of studies inspected [28,51]. In South Korea, a neighboring country of Japan, it has been reported that there are few cases of infection or disease in native amphibians, and laboratory experiments using cultured strains of the fungus have shown that native amphibians have resistance to the fungus [30]. It is likely that many Japanese amphibians have also developed this resistance through coevolution with *Bd*. It is suggested that some lineages of Caudata, such as the giant salamanders and sword tail newts, are refugia for *Bd* in the Japanese Islands.

Already, research on further *Bd* genetic diversity is continuing throughout the world, and analysis at the genome level is making great progress [40]. Based on the genome analysis to date, it is presumed that the key lineage for the origin of *Bd* exists in East and Southeast Asia [28,29]. It cannot be denied that the data presented here is only a small amount of information on DNA of a small base length and has little direct contribution to genetic analysis on a global level. Nevertheless, our results show that the Japanese archipelago is a very important area to explore *Bd* diversity in the Asian area, and that it is an area that has been left behind in *Bd* research. There are still many very interesting subjects to be discovered, such as whether the recently discovered lineage of *Bd*, *Bd*-ASIA1 and *Bd*-ASIA3, exist in the Japanese archipelago, and what is the position of *Bd* in Japan in the biogeography of *Bd* in Asia as a whole. In the future, it is necessary to restart the research project in Japan and to promote the investigation of *Bd* strains in the whole Asian region including Japan under the international collaboration system.

## Figures and Tables

**Figure 1 jof-07-00522-f001:**
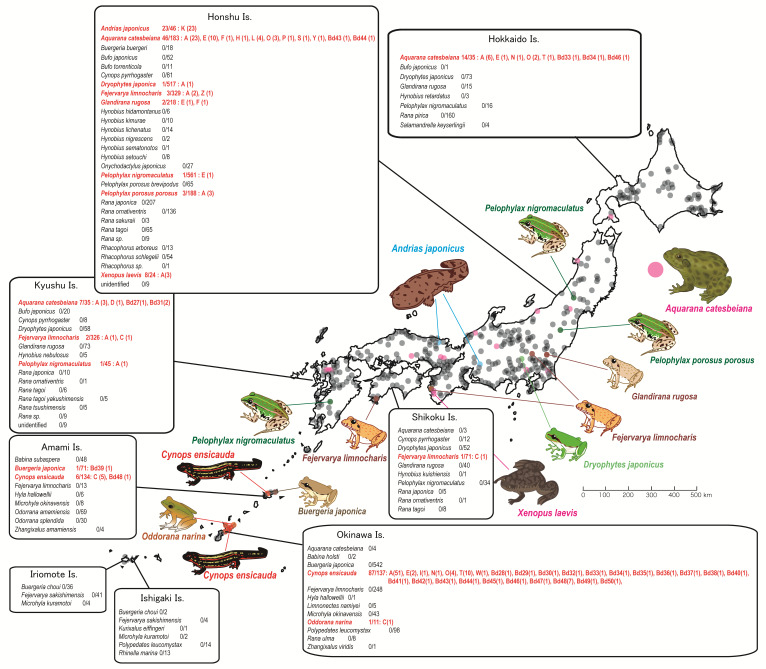
Distribution of *Batrachochytrium dendrobatidis* in amphibians across Japan. The incidence of infection in each amphibian species is defined as the number of infected animals/the number of sampled animals; the haplotype code is listed after each species. The number in parentheses represents the number of animals carrying the haplotype of interest. The circles on the map show the sampling points of swabs. Each colored circle indicates the sampling points of each amphibian species that were found to be infected.

**Figure 2 jof-07-00522-f002:**
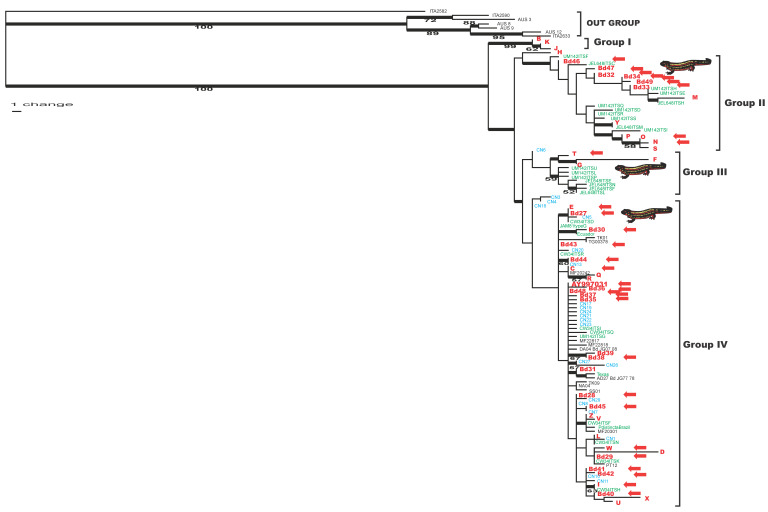
Phylogenetic tree of the ITS haplotypes of *Batrachochytrium dendrobatidis*, generated by using maximum-parsimony analysis. The numbers under the branches are the parsimony bootstrap support values for values greater than 50%. The numbers above the branches represent the lengths of the branches. Red text indicates haplotypes detected in Japan (50 types); blue text indicates sequences detected in China (21 types) [34]. Green text indicates sequences detected in North and South America (31 types) [26]; black text indicates sequences detected in the USA, Ecuador, and Italy (12 types) [32]. The red arrows indicate haplotypes of *Bd* ITS-DNA on *C. ensicauda* (30 types), indicating that all group strains of *Bd* except group I have been detected from the newt.

**Table 1 jof-07-00522-t001:** Prevalence of *Bd* infection in amphibian species in Japan.

Distribution	Species	Number of Animals Tested	Number of Animals Infected	Prevalence of Infection
Honshu Is.	*Andrias japonicus*	46	23	0.500
Honshu, Shikoku, Kyushu and South-Western Is.	*Fejervarya limnocharis*	987	6	0.006
Hokkaido, Honshu, Shikoku and Kyushu Is.	*Dryophytes japonicus*	700	1	0.001
Honshu, Shikoku, and Kyushu Is.	*Pelophylax nigromaculatus*	656	2	0.003
Honshu Is.	*Pelophylax porosus*	188	3	0.016
Honshu, Shikoku, and Kyushu Is.	*Glandirana rugosa*	346	2	0.006
Amami Is. (endemic)	*Buergeria japonica*	71	1	0.014
	*Cynops ensicauda ensicauda **	134	6	0.045
Okinawa Is. (endemic)	*Cynops ensicauda popei **	137	87	0.635
	*Odorrana narina*	11	1	0.091
Alien species	*Aquarana catesbeiana*	260	67	0.258
	*Xenopus laevis*	24	8	0.333

* Sub-species of *Cynops ensicauda*.

## Data Availability

All DNA data in this paper can be obtained from DNA databases, and their accession numbers are given in the text. Detailed collection site information for amphibians will not be released unless there is a specific reason to do so, as they include rare and endangered species.

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
