# Peer review of "Distribution and Genetic Diversity of the Amphibian Chytrid in Japan"

_jof, 2021, doi:10.3390/jof7070522_

Round 1
Reviewer 1 Report
Comments on jof-1246694 by Goka et al
This manuscript focused on the distribution and genetic diversity of amphibian Chytriodiomycosis in Japan, which is important for understanding Chytrid Fungus (Bd) as a major threat to global amphibian diversity. The manuscript is well organized and written. I only have some small comments that the authors easily deal with.
Line 40-42 in p1, “chytridiomycosis due to B. dendrobatidis (Bd) has been proposed as the infectious disease responsible for declines in wild frog populations world”. This sentence is no longer correct after many-year studies. Please specify which regions the amphibian declines caused by Bd were reported.
Line 43 in p1, please change “Scheele, B. et al. 2019” into “Scheele et al. 2019”.
After line 97 before line 98, please add a paragraph to describe the distribution of Bd in Asia by reviewing recent progress, and state that the phylogeny of Bd in Asia has been poorly studied.
Please indicate how many ITS haplotypes are used to build the phylogenetic tree in the legend of Fig 2.
Author Response
Dear Reviewer 1,
Thank you very much for the many useful suggestions.
“This manuscript focused on the distribution and genetic diversity of amphibian Chytriodiomycosis in Japan, which is important for understanding Chytrid Fungus (Bd) as a major threat to global amphibian diversity. The manuscript is well organized and written. I only have some small comments that the authors easily deal with.”
Line 40-42 in p1, “chytridiomycosis due to B. dendrobatidis (Bd) has been proposed as the infectious disease responsible for declines in wild frog populations world”. This sentence is no longer correct after many-year studies. Please specify which regions the amphibian declines caused by Bd were reported.
→We've updated the information.
Line 43 in p1, please change “Scheele, B. et al. 2019” into “Scheele et al. 2019”.
→We've corrected.
After line 97 before line 98, please add a paragraph to describe the distribution of Bd in Asia by reviewing recent progress, and state that the phylogeny of Bd in Asia has been poorly studied.
→We have described in the paragraph above that O'Hanlon et al. (2018) and Byrne et al. (2019) have analyzed and given importance to the diversity of Bd in the Asian region. We also added a sentence to the part you pointed out, “On the other hand, as mentioned above, researchers all over the world are focusing on the diversity and endemism of Bd in East and Southeast Asia, where phylogenetic studies have been lagging behind. Naturally, the Bd of the Japanese archipelago is also an important subject of investigation for clarifying the origin and diversity of Bd in Asia and the world.”
Please indicate how many ITS haplotypes are used to build the phylogenetic tree in the legend of Fig 2.
→We've added the number of haplotypes in the MP Tree in the legend.

Reviewer 2 Report
The article deals with Bd genotypes of Japan - based on a large sampling across the country. I added many suggestions in the PDF attached which will improve the ms. I think it should be publihsed and may contribute to the current knowledge of the Chytrid fungus in Japan and in a global scale too. However there is a major review to be done before it can be accepted (as indicated in the pdf).

Author Response
Author response to the Reviewer 2:
Thank you very much for the many useful suggestions.
In response to your suggestions, we have made the following changes.
TITLE
>You're not evaluating the disease, but instead the chytrid fungus - so, change it to chytrid.
→We changed the title.
ABSTRACT
>I don't agree it is not important to study Bd if you don't have declines in Japan. Discoveries in Japan may well be applied to other regions of the globe, and general knowledge of the pathogen is always good. So, please delete this entire phrase or rephrase it.
→My apologies for the misleading wording. We wanted to express that the way the government and the general public in Japan perceive Bd has changed.
We have changed the sentence as follows.
“The interest of the Japanese government and the general public in Bd has waned.”
>italics
→We have italicized the text in the Word document, but it seems that the italicization was removed during the PDF conversion process.
To editor. We would like you to confirm the above error.
>are you using italics or not for Bd?
→We would unify Bd to be italicized.
INTRODUCTION
> something is awkward in this sentence
→We've revised the text.
>you're not citing older reports of Brazil, USA and Mexico.
→We added the above references.
>4 - you're forgeting the Bd-Asia-3
→Our intention here was to document the genealogy of research on the genetic lineage of Bd. We described the discovery of Bd-Asia3 in the following paragraph. We have also rewritten this paragraph to say that three researchers discovered three major lineages other than the Bd-GPL between 2011 and 2013.
> was it the disease or the infection?
→YES, the Ceratophrys ornate developed the disease and died.
RESULTS
> this is discussion, not results
→We deleted this sentence.
> if the concept of subspecies is not applied is it still endemic?
> not sure you should use it if the general consensus is not to.
→The Herpetological Society of Japan has named the Amami Oshima and Okinawa populations of the sward tail newt as subspecies. Please refer to the following URL as well:
http://herpetology.jp/wamei/index_j.php
And Dr. Tominaga, a co-author of the present paper, also showed that the two populations are clearly genetically differentiated by molecular phylogenetic analysis, demonstrating the rationality of classifying the two populations as subspecies.
However, since it seems that the relationship between the two subspecies has not yet been recognized worldwide, we decided not to mention the subspecies relationship in the "Results" section, but to refer to the "Okinawa Island population" and the "Amami Island population" of newts, and to introduce the relationship between the subspecies in the "Discusion" section.
>this is discussion, not results.
→We deleted this sentence.
>Fig.1 and Table 1
>please, update the taxonomy - see table 1
the image quality is low, so I barrely can read the names of the species in those boxes - you must review this nomenclature as well.
→We have updated all species names.
>Table 1
> I'm not sure the concept of subspecie are being followed by recent literature - so, consider it Cynops ensicauda.
→The Herpetological Society of Japan has named the Amami Island and Okinawa Island populations of the sward tail newt as subspecies. Please refer to the following URL as well:
http://herpetology.jp/wamei/index_j.php
And Dr. Tominaga, a co-author of the present paper, also showed that the two populations are clearly genetically differentiated by molecular phylogenetic analysis, demonstrating the rationality of classifying the two populations as subspecies.
However, since it seems that the relationship between the two subspecies has not yet been recognized worldwide, we decided not to mention the subspecies relationship in the "Results" section, but to refer to the "Okinawa Island population" and the "Amami Island population" of newts, and to introduce the relationship between the subspecies in the "Discusion" section.
>why this is particular?
→As you said, there is nothing particular about this. This sentence has been deleted.
>this is discussion, not results
→We deleted this sentence.
>be careful to present results in this section, and discuss and compare your new data with others in the discussion.
→We deleted this sentence from Results, and considered to discuss about it in Discussion.
>this is discussion, not results
→We deleted this sentence from Results, and considered to discuss about it in Discussion.
>this is methods, not results
→We deleted this sentence from Results.
>I'm not sure you can call them clades, as these small fragments may generate different branch organization if you add more data... clades seems over-assertive in this case.
maybe branch?
→As you pointed out, this phylogenetic tree is only a phylogenetic inference based on a small length of sequence information, and we too thought that the term "clade" was not genetically correct.
So we replaced "clade" with "group".
>this is results mixed with discussion.
→We have rewritten this text and moved some of the content to DISCUSSION
Fig.2
> the quality of the fig is low. Maybe you could improve the text quality so it is easier to read - it is important so we could identify each branch
→The figure was drawn in Adobe Illustrator, and the resolution of the text and outline of the original drawing is considered to be acceptable.
We believe that the resolution of the figures may have been reduced during the conversion of the manuscript into a galley proof, and we are hoping that the resolution will be a little higher at the time of publication.
> I don't think this is necessary - delete the arrow and this text.
→We deleted it.
> what is the outgroup?
→We used ITS-DNA sequences from 6 other species of chytrids. We described this information in MATERIALS and METHODS.
DIACUSSION
> one very important part of the discussion is whether the genotypes you found is really linked to Bd-Brazil (and GPL). But Bd-Brazil is more important as it has been found only in the US, Korea and Brazil - so - is it possible that Bd-Brazil is also in Japan? Only in bullfrogs or also in native species? If in native species, that would be the first case of native species with Bd-Brazil outside brazil. So, explore this topic further in the discussion - since most of it was left in the results section.
→Based on the above comments, we have clearly noted that the Okinawa newts carry both Bd-GPL and Bd-Brazil, and have written a discussion on the relationship with Bd-Brazil in the world.
>have you tried to contact someone that have been sequencing Bd lineages? Maybe Tim James, Trent Garner or Matt Fisher for solving this mystery? just a suggestion.
Yes, we have been in contact with Dr. Fisher's group several times. We hope to have their cooperation when we start the project again in the future. Thank you for your kind suggestion.
>why is this relevant? think as it is one population of a single species.
→We have added scientific evidence on the subspecies relationship between the Amami and Okinawa Islands populations of the sward-tailed newt populations.
> awkward
→We revised the sentense

Round 2
Reviewer 2 Report
Dear author and editor,
The authors did a great job incorporating the previous suggestions and improved the ms. There are still some minor corrections needed that I pointed in the pdf attached.
